

# A new record of kelp *Lessonia spicata* (Suhr) Santelices in the Sub-Antarctic Channels: implications for the conservation of the "huiro negro" in the Chilean coast

Sebastián Rosenfeld[1,2], Fabio Mendez[1,3], Martha S. Calderon[1,2,4], Francisco Bahamonde[1,5], Juan Pablo Rodríguez[1], Jaime Ojeda[1,2,6], Johanna Marambio[1,2,7], Matthias Gorny[8] and Andrés Mansilla[1,2]

[1] Laboratorio de Ecosistemas Marinos Antárticos y Subantárticos, Universidad de Magallanes, Punta Arenas, Chile
[2] Instituto de Ecología y Biodiversidad, Santiago, Chile
[3] Programa de Doctorado en Ciencias Antárticas y Subantarticas de la Universidad de Magallanes, Universidad de Magallanes, Punta Arenas, Chile
[4] Instituto de Investigación para el Desarrollo Sustentable de Ceja de Selva, INDES-CES, Universidad Nacional Toribio Rodríguez de Mendoza, Chachapoyas, Peru
[5] Departamento de Ciencias y Recursos Naturales de la Facultad de Ciencias, Universidad de Magallanes, Punta Arenas, Chile
[6] School of Environmental Studies, University of Victoria, Victoria, Canada
[7] Science Faculty, Universität Bremen, Bremen, Germany
[8] OCEANA, Santiago, Chile

## ABSTRACT

The Katalalixar National Reserve (KNR) lies in an isolated marine protected area of Magellan Sub-Antarctic channels, which represent an important area for marine biodiversity and macroalgal conservation. The present study is the first report of the species *Lessonia spicata*, "huiro negro", in the Magellan Sub-Antarctic channels. This finding has implications for macroalgal biogeography and conservation concerns in the Chilean coast. In the ecological assessments of the KNR in 2018 we found populations of *L. spicata*, specifically on rocky shores of Torpedo Island and Castillo Channel. The morphological identification and molecular phylogeny based on nuclear (ITS1) sequences revealed that these populations of *Lessonia* are within the lineage of *L. spicata* of central Chile. This report increases the species richness of kelps for the Magellan Sub-Antarctic Channels from two to three confirmed species (*L. flavicans*, *L. searlesiana* and *L. spicata*), and it also extends the southern distribution range of *L. spicata*. This species has high harvest demand and is moving towards southern Chile; thus, these populations should be considered as essential for macroalgal conservation in high latitudes of South America.

## INTRODUCTION

*Lessonia* Bory (Laminariales, Phaeophyceae) is one of the most conspicuous brown macroalgal genera that inhabit the littoral to sublittoral zone of rocky coasts (~20 m depth) in temperate-cool waters of the South Pacific Ocean (*Cho et al., 2006*; *Martin & Zuccarello,*

Corresponding author
Sebastián Rosenfeld,
sebastian.rosenfeld@umag.cl

*2012*). There are currently records of 11 species of the genus *Lessonia* that are taxonomically accepted, distributed along the coasts of South America, New Zealand, Tasmania and Sub-Antarctic islands (*Cho et al., 2006*). These species have major ecological roles in the structure of benthic marine communities (*Villouta & Santelices, 1984*; *Vásquez & Santelices, 1984*), and are commercially exploited for the extraction of alginic acid (*Steneck et al., 2002*). *Lessonia* species are one of the most characteristic and abundant macroalgae (12–56°S) that inhabit the rocky shores of the Chilean coast (17–56°S) (*Searles, 1978*; *Ávila, Hoffmann & Santelices, 1985*; *Villouta & Santelices, 1986*; *Vásquez, Camus & Ojeda, 1998*; *Tellier et al., 2011*; *Martin & Zuccarello, 2012*; *Mansilla et al., 2014*). Currently, six species have been recorded in Chile: *Lessonia nigrescens* Bory, *L. berteroana* Montagne, *L. spicata* Suhr, *L. trabeculata* Villouta & Santelices, *L. searlesiana* Asensi & De Reviers and *L. flavicans* Bory (*Guiry & Guiry, 2019*). A recent morphological and molecular analysis showed that the species distributed from Peru (17°) to Puerto Montt (41°), commonly known as *L. nigrescens*, is actually two cryptic species; the populations distributed from Peru (17°S) to central Chile (30°S) correspond to *L. berteroana* Montagne, and those occurring from central Chile (29°S) to Puerto Montt (41°S) correspond to *L. spicata* (Suhr) Santelices (*González et al., 2012*; *Vega, 2016*). However, *L. nigrescens* is still a valid species, because no material of the referred species has been found near its type locality, Cape Horn.

The huiro negro kelps, which include *L. berteroana* and *L. spicata*, are heavily exploited and represent almost 70% of the kelp biomass landed annually (*Vega, Broitman & Vásquez, 2014*). This economic activity is mainly practiced in northern Chile (18–32°S), through a complex productive chain with high social impact and low added value (*Vásquez, 2008*). *L. berteroana* and *L. spicata* are exported as a natural commodity to more than 20 countries mainly due to their alginate, which has high economic value (*Westermeier et al., 2019*). Thus populations of huiro negro have economic interest along Chilean coasts, being essential to generate a stewardship from a local and large scale.

The Katalalixar National Reserve (KNR) is a national reserve area created in 1983. KNR comprises 674,500 ha and is located in a remote zone next to the village of Tortel. This area includes a wilderness temperate rainforest with a complex ecosystem of islands and fjords (*Bell, Pedersen & Newton, 2007*). The offshore area (western side) of the Magellan Sub-Antarctic Channels is one of the few places of the Magellan Biogeographic Province (MBP) (43–56°S; *Camus, 2001*) that has not been explored systematically by scientific expeditions (*Gorny & Zapata-Hernández, 2018*) (Figs. 1A and 1B). KNR is located at the southern limit of the Humboldt Current System (HCS). The HCS is a key component of the general oceanic circulation in the eastern South Pacific, being one of the most productive marine ecosystems on the earth (*Thiel et al., 2007*). The Humboldt Current System originated in southern Chile between 42 and 48°S and is characterized by a northward flow in front of South American coasts with a strong upwelling of cool nutrient-rich waters (*Silva, Rojas & Fedele, 2009*). The origin of the HCS induced a large-scale redistribution of biota, and nowadays plays a key role in the biogeography of the South Pacific (*Camus, 2001*). Thus KNR provides an enormous opportunity to

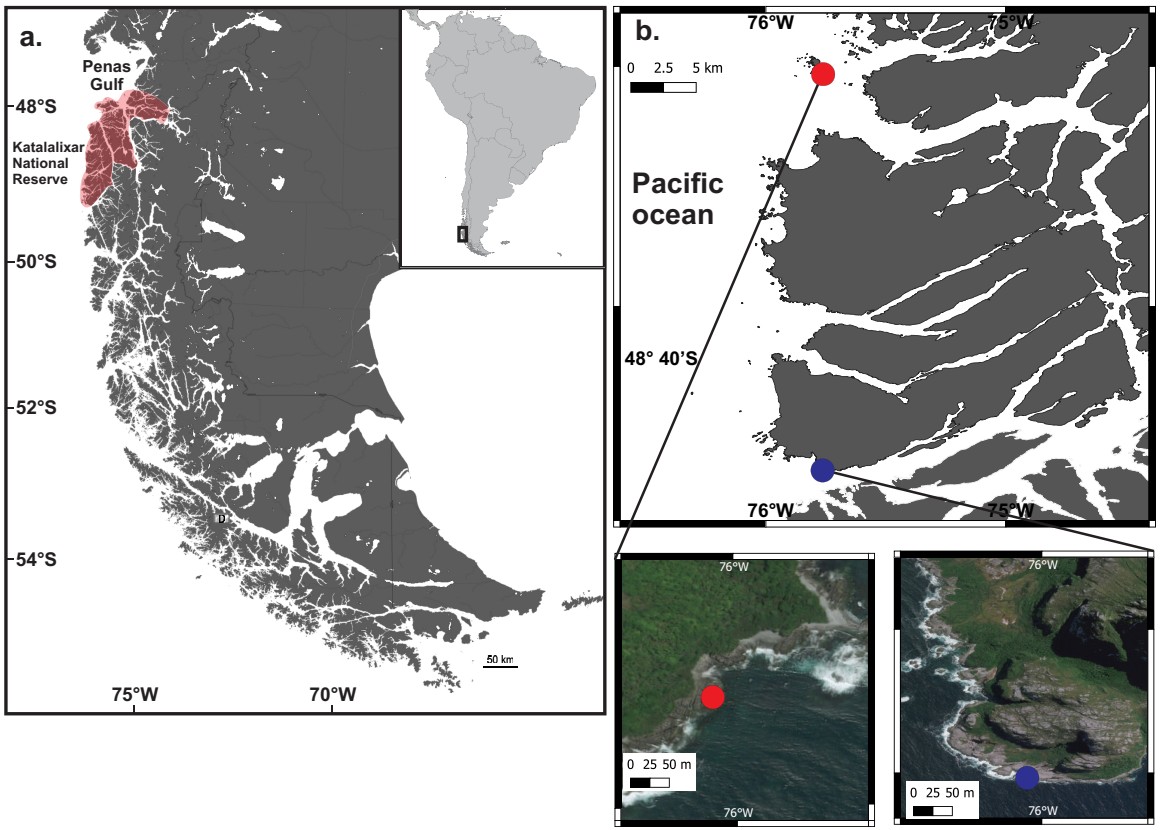

**Figure 1 Collection points of *Lessonia spicata*.** (A) Map showing the location of Katalalixar National Reserve (KNR) in central Patagonia, (B) Collection sites of *Lessonia spicata*, Torpedo Island (red circle) and Castillo Channel (blue circle) in the oceanic margin of the Campana Archipielago (KNR).

understand the taxonomic composition and biogeography of macroalgae that inhabit the southern boundary of the Humboldt Current (*Camus, 2001*; *Thiel et al., 2007*).

The present study contributes the first report of the species *L. spicata* in the Magellan Sub-Antarctic Channels. The distribution of this was thought to be limited to 41°S, but appears to be extended south of the Golfo de Penas (46° 59′–47° 40′S). Continuing survey studies will be necessary to understand the occurrence patterns of populations of *L. spicata* in the MBP.

## MATERIALS AND METHODS

Three individuals of *Lessonia spicata* were collected in the intertidal zone of Torpedo Island and Castillo Channel (Figs. 1A and 1B) in July, 2018. The specimens were air-dried and then pressed on herbarium sheets for morphological observation and molecular analysis. The Chilean Hydrographic and Oceanographic Service of the Navy (N° 13270/24/337) approved field sampling.

External and internal morphological observations were made. The anatomical observations were performed by sectioning with a razor and staining with 1% aqueous aniline blue acidified with 1% diluted HCl, and mounted in 70% glycerin. Photomicrographs were taken with a Canon Powershot S5 IS camera attached to a BX 51
Olympus microscope (Canon USA, Melville, NY, USA; Olympus Corp., Tokyo, Japan, respectively). A total of 15 replicates from the three individuals were selected for measurement of cortical cell diameter following *González et al. (2012)*; means and standard deviations were calculated. Samples of other species occurring in the Sub-Antarctic region (*L. flavicans* and *L. searlesiana*) were also analyzed for comparative purposes. Voucher specimens were deposited in the herbarium of University of Magallanes, Punta Arenas, Chile.

## Molecular analyses

Genomic DNA was extracted from ~5 mg of dried thallus ground in liquid nitrogen using a NucleoSpin Plant II Kit (Macherey-Nagel, Düren, Germany) according to the manufacturer's protocol. The PCR primers for the ITS were ITSP1-ITSRi (*Tai, Lindstrom & Saunders, 2001*; *Martin & Zuccarello, 2012*) and KP5- KG4 (*Lane et al., 2006*). Polymerase chain reaction products were purified using a NucleoSpin Gel and PCR Clean-up (Macherey-Nagel, Düren, Germany) and commercially sequenced (Macrogen, Seoul, South Korea). The electropherograms were edited using the Chromas v1.45 software (*McCarthy, 1998*) and the new generated sequences were deposited in GenBank (www.ncbi.nlm.nih.gov/genbank/).

A total of 34 ITS sequences (731 bp) were included in the construction of the phylogeny: 31 sequences belonging to the genus *Lessonia* and three outgroups, *Cymathaere triplicata* (Postels & Ruprecht) J. Agardh, *Ecklonia cava* Kjellman and *Macrocystis pyrifera* (Linnaeus) C. Agardh (Table 1). Sequences were aligned using the MUSCLE algorithm in MEGA5 v.6.06 software using the default settings (*Tamura et al., 2013*).

The phylogenic analysis was constructed using maximum likelihood (ML) and Bayesian inference (BI) analyses. The program PartitionFinder (*Lanfear et al., 2012*) were used to choose the best-fitting nucleotide substitution model under the Bayesian Information Criterion. The general time-reversible nucleotide substitution model with a gamma distribution and a proportion of invariable sites (GTR + Γ + I) was selected as the best substitution model. Maximum likelihood analysis was performed with the RAxML HPC-AVX program (*Stamatakis, 2014*) implemented in the raxmlGUI 1.3.1 interface (*Silvestro & Michalak, 2012*) with the statistical support obtained by 1,000 bootstrap replications. Bayesian inference was performed with the MrBayes v. 3.2.5 software (*Ronquist et al., 2012*) using Metropolis-coupled Markov Chain Monte Carlo ($MC^3$). The inference of Bayesian posterior probability (BPP) was inferred following *Calderon & Boo (2017)*.

The neighbor-joining analysis was performed in MEGA5 v.6.06 with the default settings software, using 1,000 bootstrap replicates.

## RESULTS

This is the first confirmed report of *L. spicata* in the Sub-Antarctic region, extending its distribution to the south by seven degrees of latitude (Fig. 2A). The sporophytes collected in the two localities have cylindrical stipes, flattened toward the beginning of the blades, with a regular, almost dichotomous long lanceolate blade with a spike (Figs. 2B–2E).

**Table 1 List of species used in DNA analyses, information on collections and accession numbers in GenBank (sequences generated in the present study are shown in bold).**

| Species | Collection site | Voucher code | ITS |
|---|---|---|---|
| **LesA** | Torpedo island, Aysen, Chile | | MN061669 |
| **LesB** | Channel Castillo, Aysen Chile | | MN061670 |
| **LesC** | Channel Castillo, Aysen, Chile | | MN061671 |
| *Lessonia adamsiae* | South Promontory, The Snares, New Zealand | A626 | GU593802[1] |
| *Lessonia adamsiae* | Tahi, The Snares, New Zealand | A614 | GU593799[1] |
| *Lessonia berteroana* (as *L. nigrescens* northern lineage) | San Marcos, Tarapaca, Chile | B858 | GU593781[1] |
| *Lessonia berteroana* (as *L. nigrescens* northern lineage) | San Marcos, Tarapaca, Chile | B859 | GU593782[1] |
| *Lessonia brevifolia* | Smoothwater Bay, Campbell Is., New Zealand | A548 | GU593803[1] |
| *Lessonia brevifolia* | Antipodes, New Zealand | A973 | GU593804[1] |
| *Lessonia brevifolia* | Perseverance Harbour, Campbell Is., New Zealand | B296 | GU593805[1] |
| *Lessonia corrugata* | Gov. Is. Reserve, Tasmania, Australia | | AY857902[2] |
| *Lessonia corrugata* | Bicheno, Tasmania, Australia | A985 | GU593794[1] |
| *Lessonia corrugata* | Skeleton Pt., Tasmania, Australia | C057 | GU593795[1] |
| *Lessonia flavicans* | Rookery Bay, East Falkland, Falkland Islands | A634 | GU593786[1] |
| *Lessonia flavicans* (as *Lessonia vadosa*) | Punta Arenas, Patagonia, Chile | B985 | GU593789[1] |
| *Lessonia spicata* (as *L. nigrescens* IA lineage) | La Pampilla, Coquimbo, Chile | A581 | GU593775[1] |
| *Lessonia spicata* (as *L. nigrescens* IA lineage) | Bahia Mansa, Osorno, Chile | B719 | GU593780[1] |
| *Lessonia tholiformis* | Wharf reef, Owenga, Chatham Is, New Zealand | A518 | GU593797[1] |
| *Lessonia tholiformis* | Wharekauri, Chatham Is, New Zealand | A532 | GU593798[1] |
| *Lessonia trabeculata* | Punihuil, Chiloe Is, Chile | B715 | GU593783[1] |
| *Lessonia trabeculata* | Punihuil, Chiloe Is, Chile | B716 | GU593784[1] |
| *Lessonia variegata* (as *L. variegata* lineage N) | North Cape, Northland, New Zealand | A557 | GU593808[1] |
| *Lessonia variegata* (as *L. variegata* lineage N) | Maitai Bay, Northland, New Zealand | B129 | GU593809[1] |
| *Lessonia variegata* (as *L. variegata* lineage N) | The Sailors Grave, Coromandel, New Zealand | B312 | GU593810[1] |
| *Lessonia variegata* (as *L. variegata* lineage K) | South Bay, Kaikoura, New Zealand | A138 | GU593817[1] |
| *Lessonia variegata* (as *L. variegata* lineage K) | New Wharf, Kaikoura, New Zealand | A606 | GU593818[1] |
| *Lessonia variegata* (as *L. variegata* lineage S) | Curio Bay, Catlins, New Zealand | A434 | GU593820[1] |
| *Lessonia variegata* (as *L. variegata* lineage S) | Causet Cove, Doubtful Sound, New Zealand | C154 | GU593821[1] |
| *Lessonia variegata* (as *L. variegata* lineage W) | Princess Bay, Wellington, New Zealand | A001 | GU593811[1] |
| *Lessonia variegata* (as *L. variegata* lineage W) | Cape Palliser, Wairarapa, New Zealand | A613 | GU593815[1] |
| *Lessonia variegata* (as *L. variegata* lineage W) | Riversdale Beach, Wairarapa, New Zealand | A025 | GU593816[1] |
| *Cymathaere triplicata* | Whiffen Spit, Sooke, BC, Canada | | AY857884[2] |
| *Macrocystis pyrifera* | California, USA | | AF319037[3] |
| *Ecklonia cava* | | | GU593773[1] |

**Notes:**
[1] *Martin & Zuccarello (2012).*
[2] *Lane et al. (2006).*
[3] *Yoon et al. (2001).*

## Internal anatomy

Our specimens showed several layers of cortical tissue with cells of smaller diameter compared to *L. searlesiana* (Figs. 3B, 3E, and 3H) and *L. flavicans* (Figs. 3C, 3F, and 3I), moreover no lacunas were observed in our samples, unlike *L. flavicans* (Figs. 3C and 3H).

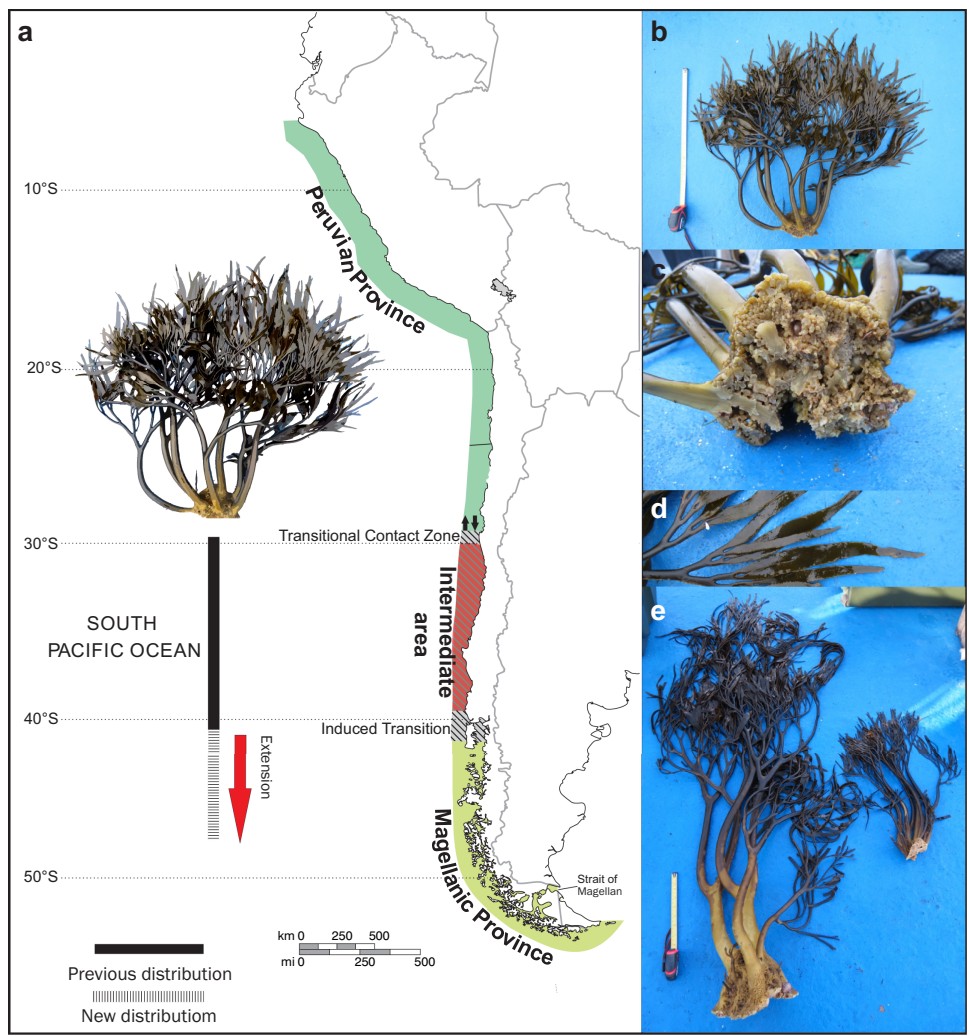

**Figure 2 Distribution of *Lessonia spicata* (interspersed bars), showing its previously known distribution (solid bars).** (A) Habitat of specimen collected in both sites (Torpedo Island and Castillo Channel). We included the Chilean biogeographical classification of *Camus (2001)*. (B) Habitat of specimen collected in Torpedo Island (LMS000001). (C) Discoid holdfasts of specimen collected in Torpedo Island (LMS000001), (D) Blades of specimen collected in Torpedo Island (LMS000001), (E) Habitat of specimen collected in Castillo Channel (LMS000002, LMS000003).

The medulla was composed of elongated medullary cells with filamentous elements (Fig. 3G). The internal anatomy was composed of a narrow cortex (Fig. 3A), with cortical cell diameter of 25.91 ± 2.90 for the individual 1, 28.22 ± 2.10 for individual 2 and 27.02 ± 2.27 for the individual 3 (Table 2).

## Phylogenetic analysis

The ITS phylogeny placed our specimens within the lineage of *L. spicata* of central Chile (Fig. 4A). The phylogenetic trees constructed by ML and BI had the same topology except for the phylogenetic position of *L. corrugata* and *L. variegata* from northeastern South Is. The three specimens analyzed consistently formed a strongly supported clade with

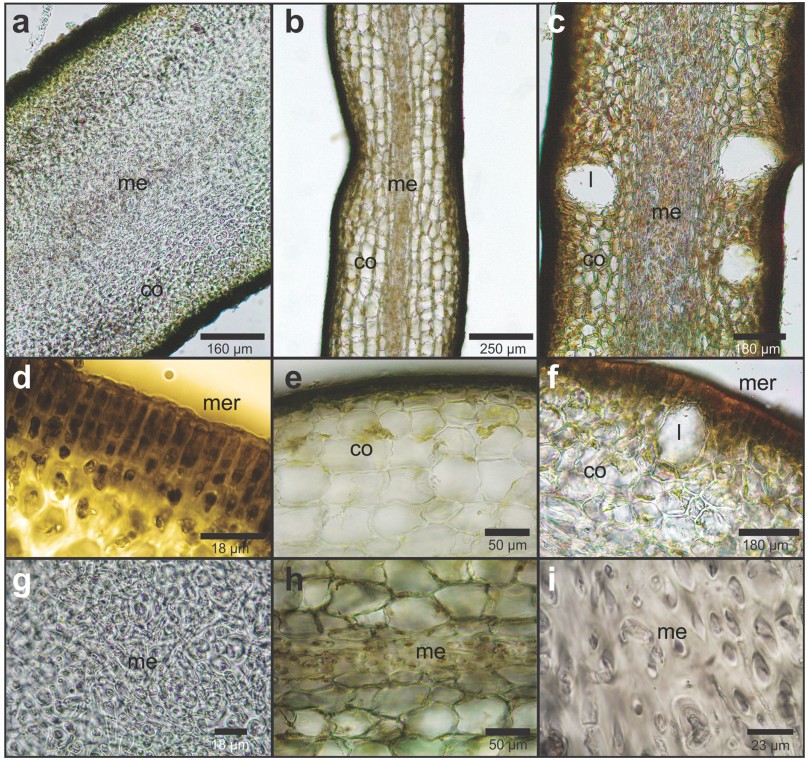

**Figure 3 Cross section of the medial part of mature fronds of *Lessonia* species who inhabit the Sub-Antarctic channels.** Cross section of the medial part of mature fronds of *Lessonia spicata* collected in the Katalalixar Reserve (A, D and G), of *L. searlesiana* from Fuerte Bulnes (B, E and H) and *L. flavicans* from Horn Island (C, F and I); mer = meristoderm, co = cortex and me = medulla, l = lacuna.

**Table 2 Morphological measurements (mean ± SE) of individuals collected in Torpedo Island and Castillo Channel.**

| | | External measurements | | | Internal measurements | |
|---|---|---|---|---|---|---|
| | | TL | DD | NS | DC | |
| Individual | Site | | | | | |
| 1 | Torpedo Island | 68 | 10 | 9 | 25.91 ± 2.90 | This study |
| 2 | Castillo Channel | 166 | 21 | 6 | 28.22 ± 2.10 | This study |
| 3 | Castillo Channel | 55 | 5 | 13 | 27.02 ± 2.27 | This study |
| Average | | 96.33 ± 60.68 | 12 ± 8.19 | 9.33 ± 3.51 | 27.05 ± 1.15 | This study |
| *L. spicata* | | | | | | |
| | Maintencillo | 150 ± 13.3 | – | – | 25.7 ± 1.4 | (*González et al., 2012*) |
| | Matanzas | 160 ± 5.0 | – | – | 27 ± 1.6 | (*González et al., 2012*) |
| | Calfuco | 120 ± 7.2 | – | – | 30 ± 1.8 | (*González et al., 2012*) |

**Note:**
External morphological data: TL, thallus length (cm); DD, disc diameter; NS, number of stipes. Internal morphological data: DC, diameter of cortical cells.

sequences of *L. spicata* (97% for ML and 0.96 for BPP) collected in Chile; having to *L. berteroana* and *L. trabeculata* as sister taxa. The cladogram was consistent with the phylogenetic tree (Fig. 4B). Variable sites occurred at 201 positions (27.5%), and 123

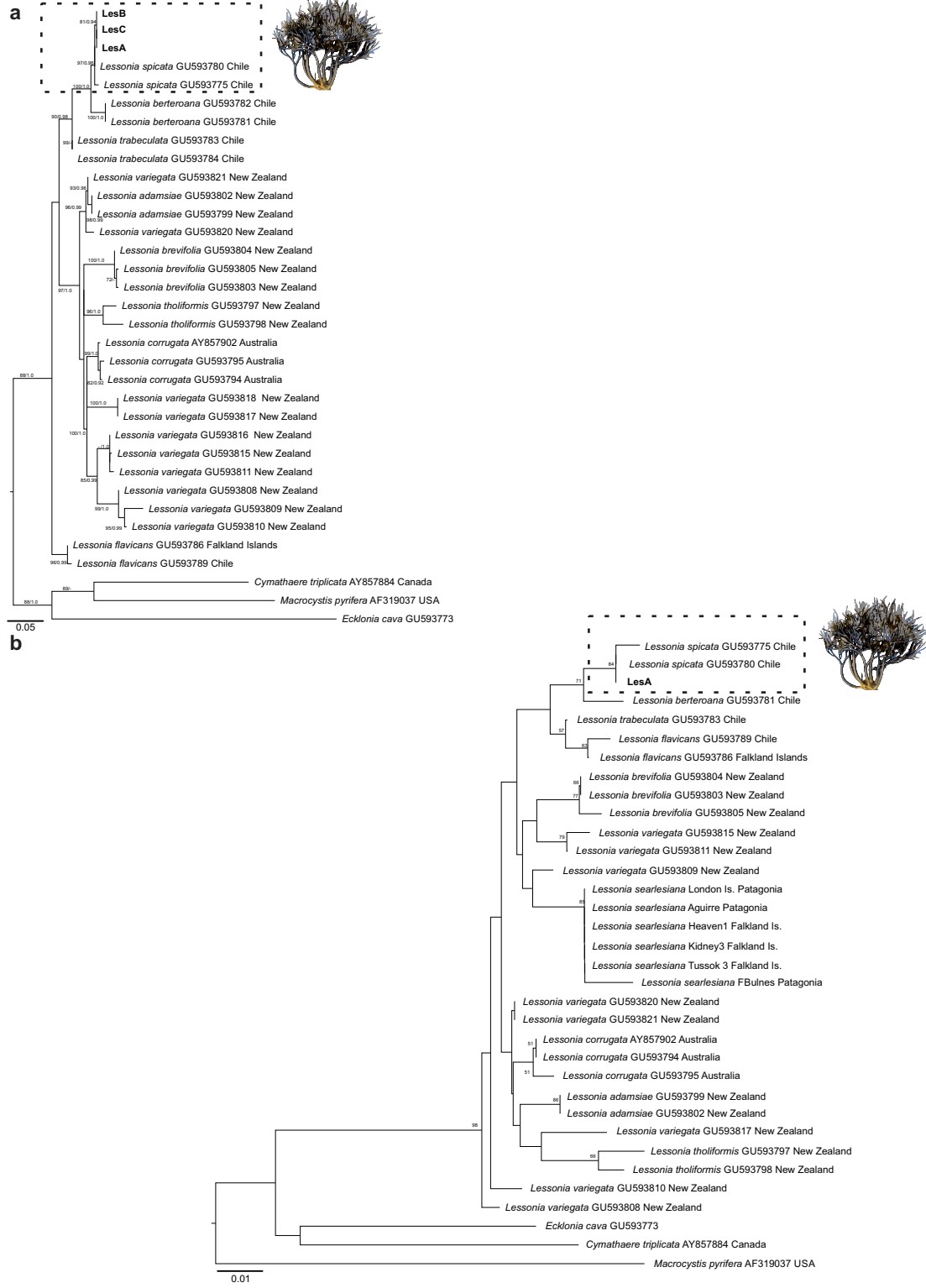

**Figure 4 Phylogenetic tree of ITS sequences obtained by maximum likelihood (ML) inference.** (A) Phylogenetic tree of ITS sequences obtained by maximum likelihood (ML) inference. ML bootstrap values (≥50%) and Bayesian posterior probabilities (≥0.90) are indicated next to branches. (B) Cladograms of ITS sequences obtained by the neighbor joining (NJ) method. Bootstrap values (≥50%) are indicated next to branches. The sequence for taxa in bold was generated in this study.

positions (16.8%) were parsimoniously informative. Intraspecific divergence of *L. spicata* from three different populations ranged between 0.0% and 0.2% (0–3 bp). *L. spicata* differed by 0.8–1.0% from *L. berteroana* and by 1.1–1.3% from *L. trabeculata*. *L. variegata* is a non-monophyletic species complex of four different species.

## DISCUSSION

We confirm here the presence of *L. spicata* both morphologically and genetically, whose individuals correspond to the central Chile lineage described by *González et al. (2012)*. Morphologically these features correspond to those described for *L. spicata* by *von Suhr (1839)* and *González et al. (2012)*. These values also agree with those mentioned by *González et al. (2012)* for *L. spicata*. Genetically our phylogeny is consistent with those of previous studies that show *Lessonia* as a monophyletic lineage (*Lane et al., 2006*, *Martin & Zuccarello, 2012*).

*Lessonia* species are a characteristic component of benthic ecosystems in this region (*Searles, 1978*; *Martin & Zuccarello, 2012*). We highlight two aspects about the importance of this report of *L. spicata* for this area: (a) we increase the knowledge of the species richness of kelps for the Sub-Antarctic Channels, and (b) this species has a strong extraction activity which we hypothesize that will move southward in the near future, therefore these populations should be properly preserved in order to prevent high risk of human impact.

The name *L. spicata* was proposed because it was the oldest name available to assign the lineage of central Chile, populations between 29° and 43°S. However, *L. spicata* would be a provisional name mainly because no representative specimens of *L. nigrescens* have been found near the type locality Cape Horn. Therefore, if the true *L. nigrescens* belongs to one of the lineages already described or to a new one, this name would have priority (*González et al., 2012*). In the MBP *L. nigrescens* has been recorded not only for Cape Horn; *Searles (1978)* reported a population in the Trinidad Channel (Puerto Alert 49° 53.6 ′S), and two others in the Aysén region (*Searles, 1978*). Puerto Alert is 126 km south of Castillo Channel where we found the population of *L. spicata*. Therefore, it is likely that Searles' records (1978) correspond to populations of *L. spicata*. Finally, it is important to mention that, like *González et al. (2012)*, in recent expeditions to the Diego Ramirez and Cape Horn archipelago—which are related to the characterization of the Diego Ramírez-Drake Passage Marine Park (*Rozzi et al., 2017*)—we have not found populations of *L. nigrescens*, only individuals of *L. flavicans* (*Rozzi et al., 2017*). Therefore, in the absence of biological material from the type locality the status of *L. nigrescens* is still in doubt, and the lineage of central Chile that now extends south of 43°S should continue to be named as *L. spicata*.

Several bio-geographical breaks have been described along the coast of Chile (*Santelices & Meneses, 2000*; *Tellier et al., 2009*; *Fraser et al., 2010*); one of the most relevant for many taxa is at 42°S (*Brattström & Johanssen, 1983*; *Lancellotti & Vásquez, 1999*; *Valdovinos, Navarrete & Marquet, 2003*). For macroalgae and particularly for kelp species such as *Durvillaea antarctica*, a marked divergence is present south of 43°S, where populations between 49 and 55°S are genetically different from the rest of the populations

occurring in the Chilean coast (32 and 43°S) (*Fraser et al., 2010*). These authors suggested that although *D. antarctica* has a high dispersion capacity due to its buoyancy (rafting), it could only colonize free coasts, since it would have limited potential to increase gene flow between established populations. Therefore, it is interesting that although *L. spicata* has a low-dispersal capacity in comparison to *D. antarctica* (*Oppliger et al., 2012*), since it does not have the buoyancy capacity, there is a single genetic unit in the individuals collected in this study and individuals from the central zone of Chile. *L. spicata* must have some physiological adaptations which allowed it to colonize and inhabit areas of high latitudes. In this sense, this species has been described as a perennial seaweed and has not been found in the "bank of microscopic forms" in the Chilean central coast (boulders and water from tidal pools) (*Santelices et al., 1995*; *Santelices, Aedo & Hoffmann, 2002*). However, it has been observed that microscopic form of *L. spicata* can survive up to 90 days in total darkness and propagules can germinate in total absence of light (*Santelices, Aedo & Hoffmann, 2002*). This high capacity for tolerance to darkness could be a key strategy to colonize new areas with a significant seasonal changes in daylight hours and luminosity (Photosynthetically Active Radiation) during the winter period (*Ojeda et al., 2019*). Nevertheless, future studies and a greater number of samples along the Chilean coast (mainly the area between 41 and 48°S) will help to elucidate its biogeographic history and how much structure and connectivity the populations of *L. spicata* present throughout their distribution (29–48°S).

The harvesting pressure on the genus *Lessonia* has increased alarmingly along the Chilean coast, so we should take a precautionary approach to potential harvesting of *L. spicata* in its austral distribution range. *L. berteroana* (sister species of *L. spicata*) is currently the most exploited seaweed in South America; the main landings are in northern Chile (*Westermeier et al., 2019*). *Lessonia* is socially important in this region because many artisanal fishers depend directly or indirectly on its harvest (*Vega, Broitman & Vásquez, 2014*). However, high demand, lack of oversight and harvest methods have created a concerning scenario for kelp forests (*Vega, Broitman & Vásquez, 2014*; *Westermeier et al., 2019*). The extraction of *L. spicata* in southern Chile began in 2012, and its extractive pressure has been moving southward, mainly between 33 and 41°S (*SERNAPESCA, 2019*). In the Chilean Los Lagos Region (41°S), between 2014 and 2017 landing increased from 494 to 747 dry tons of *L. spicata* (*SERNAPESCA, 2019*). This gradual increase should draw attention to kelp forest conservation, since there is evidence on sustainability problems that *Lessonia* populations have experimented and their biodiversity in northern Chile (*Vega, Asorey & Piaget, 2016*). This concern acquires significant relevance if we consider that the Magellan Sub-Antarctic Channels are the austral distribution range of *L. spicata*, where kelp forest populations are important for sustainability of small-scale fisheries (e.g., king crab; *Cárdenas, Cañete & Mansilla, 2007*), indigenous traditions (*Ojeda et al., 2018*) and terrestrial and marine biodiversity (*Darwin, 1839*; *Rosenfeld et al., 2014*).

## CONCLUSION

Despite the geographical distance and the presence of important biogeographic breaks (41 and 46°S), our results confirm that the individuals collected in the coastal zone of the

Katalalixar Reserve are the species *L. spicata*. The strong morphological and genetic evidence are indicating that the individuals analyzed are associated with the lineage of central Chile, and the populations of *L. spicata* would inhabit the area exposed to the Pacific.

With diverse industrial uses, including providing phycocolloids in the form of alginate *L. spicata* is a potentially important economic resource in the Chilean coast. However, with extractive pressure moving to the south, caution is needed given that this kelp serves not only as a habitat for many animals but also as a spawning ground for some benthic (e.g., gastropods) species.

## ACKNOWLEDGEMENTS

Sebastián Rosenfeld, Jaime Ojeda, Johanna Marambio and Andrés Mansilla want to thank the program "CONICYT PIA SUPPORT CCTE AFB170008" The authors acknowledge the invitation to participate in the KATALALIXAR Expedition (winter 2018) supported by OCEANA Chile. We especially thank Lafayette Eaton for English revision and editing and to Alonso Vega, Blanca Figuerola and one anonymous reviewer for their helpful comments.

### Funding

This research was funded by Chile's National Council for Research in Science and Technology (CONICYT) FONDECYT Program grant 1180433 to Andrés Mansilla, and the CONICYT-FB001 FONDECYT 3180539 project to Martha S. Calderon. The expedition to the Katalalixar National Reserve was financed by OCEANA Chile. The funders had no role in study design, data collection and analysis, decision to publish, or preparation of the manuscript.

### Grant Disclosures

The following grant information was disclosed by the authors:
Chile's National Council for Research in Science and Technology (CONICYT) FONDECYT Program grant 1180433 and the CONICYT-FB001 FONDECYT 3180539 project to Martha S. Calderon.
The expedition to the Katalalixar National Reserve was financed by OCEANA Chile.

### Competing Interests

The authors declare that they have no competing interests.

### Author Contributions

- Sebastián Rosenfeld conceived and designed the experiments, performed the experiments, analyzed the data, contributed reagents/materials/analysis tools, prepared figures and/or tables, authored or reviewed drafts of the paper, approved the final draft, identify the species in the field, and collect it.

- Fabio Mendez conceived and designed the experiments, contributed reagents/materials/analysis tools, approved the final draft, help in the expedition.
- Martha S. Calderon conceived and designed the experiments, performed the experiments, analyzed the data, contributed reagents/materials/analysis tools, authored or reviewed drafts of the paper, approved the final draft.
- Francisco Bahamonde conceived and designed the experiments, performed the experiments, contributed reagents/materials/analysis tools, prepared figures and/or tables, approved the final draft.
- Juan Pablo Rodríguez conceived and designed the experiments, contributed reagents/materials/analysis tools, prepared figures and/or tables, approved the final draft.
- Jaime Ojeda conceived and designed the experiments, contributed reagents/materials/analysis tools, prepared figures and/or tables, authored or reviewed drafts of the paper, approved the final draft.
- Johanna Marambio conceived and designed the experiments, contributed reagents/materials/analysis tools, prepared figures and/or tables, authored or reviewed drafts of the paper, approved the final draft.
- Matthias Gorny conceived and designed the experiments, contributed reagents/materials/analysis tools, approved the final draft, help in the expedition.
- Andrés Mansilla conceived and designed the experiments, contributed reagents/materials/analysis tools, approved the final draft, he contributed with the funding.

## Field Study Permissions

The following information was supplied relating to field study approvals (i.e., approving body and any reference numbers):

The field samplings were approved by the Chilean Hydrographic and Oceanographic Service of the Navy (SHOA). This species is not protected by the Chilean Fishery Subsecretary and has not been included in the Chilean fishery statistics in the Aysen Region. The permission to undertake field studies and to collect specimens was issued by the Chilean Hydrographic and Oceanographic Service of the Navy Sub-Director (Felipe Barrios Burnett), under the technical memorandum N° 13270/24/337.

## DNA Deposition

The following information was supplied regarding the deposition of DNA sequences:

The *Lessonia spicata* ITS sequences are accessible via GenBank: LesA MN061669, LesB MN061670, LesC MN061671.

## Data Availability

The detailed information on the new records of *Lessonia spicata* are available in File S1.

## Supplemental Information

Supplemental information for this article can be found online at http://dx.doi.org/10.7717/peerj.7610#supplemental-information.

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
