# Peer review of "A new record of kelp Lessonia spicata (Suhr) Santelices in the Sub-Antarctic Channels: implications for the conservation of the “huiro negro” in the Chilean coast"

_PeerJ, doi:10.7717/peerj.7610_

## Round 0.1 · original submission · Major Revisions

Dear Sebastián Rosenfeld,

Thank you for submitting your manuscript to PeerJ. Your manuscript entitled “A new record of kelp Lessonia spicata (Suhr) Santelices in the Sub-Antarctic Channels: implications for the conservation of the “huiro negro” in the Chilean coast” has been evaluated by two peer reviewers, and the reviewer comments are appended below.

Reviewer #1 considers the data in this manuscript are valuable and the macroalgal taxonomy and biodiversity research will highly benefit from this study’s discoveries. However, the manuscript needs minor revisions and thorough proofreading. Reviewer #2 thinks the topic is interesting for the conservation and management of the kelp species although the manuscript is currently rather descriptive than being written as a hypothesis driven study. However, he/she has raised points that need to be addressed such as a review of all the citations and references in the manuscript, a description of the results of the comparative analysis between Lessonia species and moving part of the discussion to the results section.

Based on the referees' recommendations, I arrive at this decision: The manuscript does merit publication in PeerJ but it is not acceptable in its current form and needs a major revision based on the reviews. I therefore invite you to resubmit a revised version, taking into account all the points raised with a special focus on problems related to the literature and results and discussion sections. Please carefully consider the comments of the reviewers and provide a point-by-point response which clearly defines the changes made.

Please also carefully proofread the manuscript to eliminate any typographical or grammatical errors (e.g. line 76: delete the space between (MBP) and (43°-56°S; line 81: delete “in the”).

Thank you for your patience with the evaluation process and for choosing PeerJ.

I look forward to receiving your revised manuscript.

Yours sincerely,


Blanca Figuerola
* * *
Academic editor
PeerJ
* * *
Reviewer 1 ·

Basic reporting

no comment

Experimental design

no comment

Validity of the findings

no comment

Additional comments

The macroalgal community has been strangling for years with the taxonomy of Lessonia species for this region. The macroalgal taxonomy and biodiversity research will highly benefit from this study’s discoveries.

However, the manuscript needs minor revisions and thorough proofreading. Below are highlighted the points to be addressed.

• The word “macroalga” is mentioned through the text but the majority of the studies use the word “macroalgal”, e.g. Line 24 and Line 34: “macroalga” conservation should be “macroalgal” conservation, and Line 41: “macroalga” genera should be “macroalgal” genera.

• References format throughout the manuscript should be checked again and make sure they all are consistent, e.g. Line 46: et al. is not italicized even though it is in the rest of the manuscript, and Line 54: a comma is missing before the date.

• Line 55: It would be better if “17° to 41°” are mentioned with locations as in the lines 57 and 58, e.g. “Peru (17°S) to Peru (17°S)”, for better understanding.

• Line 68: Since when is the Katalalixar National Reserve a marine protected area (MPA)? Some info about how long is this MPA running (since when) would be useful.

• Line 75: It would be beneficial for the reader if you briefly mention what is the Humboldt Current and why is it important for the taxonomic composition and biogeography.

• Line 76-79: Fig. 1a could be mentioned along those lines.

• Line 162: It is more accurate to say “we increase the knowledge of the species richness” instead of “we increase the species richness”.

• Line 164: It is wiser to say, “we hypothesize that” or “we predict that” … “will move southward in the near future”.

• Line 224: “spawning ground” is a more accurate term.

·

Basic reporting

It is an exploratory study, without hypothesis, that reports the presence of a kelp species (Lessonia spicata) in the sub-Antarctic Magellanic channels using morphological and phylogenetic analysis. The manuscript is written with appropriate technical English.
The theoretical justification of the study is adequate. The bibliographical references are appropriate. I suggest to review all the literature cited in the manuscript. For example: "Vega et al. 2016 "(line 207), and" Suhr 1839 "(line 148) has no bibliographical reference. The reference "Fraser et al. 2009 "(line 260) is not cited in the manuscript. The reference "McCarthy C. 1998" on line 294 is incomplete. "Sernapesca 2019" (line 203 and 205) in capital letters. The year of some references is in parentheses (line 357 and 360). "Yoon et al. 2001 "in the legend of Table 1 has no reference.
The structure of the article is adjusted to the format of the PeerJ. The figures and tables are clear and have appropriate legends. Raw data is available and accessible. In the legend of figure 4, it is necessary to differentiate the text of the phylogenetic tree (a) from the cladograms of ITS sequences (b).

Experimental design

no comment

Validity of the findings

Results.
In the text of the manuscript are the same data presented in table 2 (lines 132-135). In addition, there is an error in the holdfast diameter of a individuals collected in the Castillo Channel (9 cm in the text and 5 cm in the table). I suggest first describing the results of the comparative analysis between Lessonia species (figure 3, see line 150-154 discussion); Then, I suggest complementing the analysis with the results of table 2 (see line 135 and 136, not including average measurements).
In molecular analysis, I suggest first describing the results of the phylogenetic tree (figure 4a) and then the results of the cladogram (figure 4b). Why in the tree of figure 4b does only LesA appear?

Discussion.
I suggest moving part of the first paragraph of the discussion (line 150 to line 157) has results - morphological analysis. Another sentence of this paragraph could be part of the results of the phylogenetic analysis: "Intraspecific divergence of L. spicata ..." (Lines 155-157).
I also suggest moving (or copying) “This is first confirmed report of L. spicata…” (line 159-160) to the beginning of results (line 130). Leave "Fig2a" only in results.
I suggest briefly discussing "the bank of algal microscopic forms" (Hoffman & Santelices 1991 MEPS 79: 185-194) as a potential mechanism of population persistence of L. spicata in high latitudes (see lines 189-194).

Additional comments

The recording of Lessonia spicata in the sub-Antarctic channels is an important finding for the conservation and management of the kelp species of the South American Pacific. In this context, the justification of the problem in the manuscript is well established and the methods are adequate.
However, the results should describe the figures and tables of the morphological and phylogenetic analyzes with concluding sentences; following the numerical order of figures and tables. In this context, in the discussion there are phrases that should be in the results section. In addition, citations and references should be reviewed in depth.
Finally, it would be very interesting to briefly discuss the hypothesis of the "bank of algal microscopic forms" as a potential mechanism of persistence of populations of L. spicata in high latitudes. What would be the eventual effect on the phylogeny?

---

## Round 0.2 · accepted · Accept

Dear Dr. Rosenfeld

I am pleased to inform you that your paper has been accepted for publication without further changes.

Thank you for submitting your work to PeerJ. We hope you consider us again for future submissions.

Best regards,

Blanca Figuerola
Academic Editor, PeerJ